# *Chlamydomonas reinhardtii*, an Algal Model in the Nitrogen Cycle

**DOI:** 10.3390/plants9070903

**Published:** 2020-07-16

**Authors:** Carmen M. Bellido-Pedraza, Victoria Calatrava, Emanuel Sanz-Luque, Manuel Tejada-Jiménez, Ángel Llamas, Maxence Plouviez, Benoit Guieysse, Emilio Fernández, Aurora Galván

**Affiliations:** 1Departamento de Bioquímica y Biología Molecular, Campus de Rabanales y Campus Internacional de Excelencia Agroalimentario (CeiA3), Edif. Severo Ochoa, Universidad de Córdoba, 14071 Córdoba, Spain; b22bepec@uco.es (C.M.B.-P.); b82capom@uco.es (V.C.); q92salue@uco.es (E.S.-L.); q62tejim@uco.es (M.T.-J.); bb2llaza@uco.es (Á.L.); 2School of Food and Advanced Technology, Massey University, Private Bag, 11222 Palmerston North, New Zealand; M.Plouviez@massey.ac.nz (M.P.); B.J.Guieysse@massey.ac.nz (B.G.)

**Keywords:** algae, *Chlamydomonas*, nitrogen cycle, nitric oxide, nitrous oxide

## Abstract

Nitrogen (N) is an essential constituent of all living organisms and the main limiting macronutrient. Even when dinitrogen gas is the most abundant form of N, it can only be used by fixing bacteria but is inaccessible to most organisms, algae among them. Algae preferentially use ammonium (NH_4_^+^) and nitrate (NO_3_^−^) for growth, and the reactions for their conversion into amino acids (N assimilation) constitute an important part of the nitrogen cycle by primary producers. Recently, it was claimed that algae are also involved in denitrification, because of the production of nitric oxide (NO), a signal molecule, which is also a substrate of NO reductases to produce nitrous oxide (N_2_O), a potent greenhouse gas. This review is focused on the microalga *Chlamydomonas reinhardtii* as an algal model and its participation in different reactions of the N cycle. Emphasis will be paid to new actors, such as putative genes involved in NO and N_2_O production and their occurrence in other algae genomes. Furthermore, algae/bacteria mutualism will be considered in terms of expanding the N cycle to ammonification and N fixation, which are based on the exchange of carbon and nitrogen between the two organisms.

## 1. Introduction

Nitrogen (N) is an essential macronutrient that supports life in all living beings. This nutrient is an elemental constituent of biomolecules, such as nucleic acids, proteins, chlorophylls, cofactors, and signal molecules, among others. 

In nature, living organisms can use different N reservoirs, and the biotic and abiotic conversions among different N forms constitute the biogeochemical N cycling. These N reservoirs and N flux in terrestrial and marine ecosystems have been recently reviewed [1]. Ammonium (NH_4_^+^) bound into rocks and sediments is the largest N reservoir (1.8 × 10^10^ Tg nitrogen). However, this NH_4_^+^ has a minimal impact on the annual N cycling because it is accessible only upon erosion [1]. The largest available N reservoir for organisms is dinitrogen gas (N_2_) with 3.9 × 10^9^ Tg, which represents 79% of the atmospheric air. However, due to the reduced reactivity of the triple bonded N_2_, only a small group of organisms having the dinitrogenase enzyme complex can use it as N source [1,2,3]. The global and usable N-reservoirs are organic nitrogen (9 × 10^5^ Tg), nitrate (NO_3_^−^) (6 × 10^5^ Tg), and nitrous oxide (N_2_O) (2000 Tg) [1,2,3,4]. The ammonia (NH_3_) reservoir estimated in the marine ecosystem is 340–3600 Tg, but it is unknown in the terrestrial ecosystem [1]. Other global inorganic N sources such as nitrite and nitric oxide are considered minor reservoirs.

N compounds exhibit different oxidation states ranging from −3 (organic N and NH_3_/NH_4_^+^) to +5 (NO_3_^−^), with intermediate values in compounds such as hydrazine (N_2_H_4_) (−2), hydroxylamine (NH_2_OH) (−1), N_2_ (0), nitrous oxide (N_2_O) (+1), NO (+2), nitrite, NO_2_^−^ (+3), and nitrogen dioxide (NO_2_) (+4). Bacteria can catalyze many redox reactions involving the above compounds, which can be used to provide nitrogen for growth and redox energy to fuel cells. Bacterial species have been classically classified according to the reactions that they mediate within the N cycle (nitrification, denitrification, fixation, ammonification, anammox, assimilatory, and dissimilatory processes). Notwithstanding, DNA genome sequencing has revealed the set of genes present in each microorganism and highlighted the enormous metabolic flexibility of some of the organisms studied, making it almost impossible to classify them using only the classical divisions [1,2]. Moreover, it is expected that further research will unravel new N-transforming reactions and enzymes that will emphasize the metabolic flexibility of many of these organisms [1]. 

Prokaryotic organisms exhibit an enormous versatility to metabolize most of the N compounds, and they are the main players for most of the transformations in the N cycle. In addition, enzymes required for N_2_ fixation, such as nitrogenase, are exclusively found in prokaryotes or bacteria involved in symbiosis/mutualism with other eukaryotic organisms. For instance, N_2_-fixing bacteria from the order Rhizobiales establish symbiotic interactions with legume plants, whose crops provide almost 20% of food protein in the world [5]. Other examples are found in marine ecosystems, where a symbiotic interaction between the haptophyte algae and the most widespread N_2_-fixing cyanobacterium ‘*Candidatus*
*Atelocyanobacterium* thalassa’ (UCYN-A) has been described [6,7]. In the lab, the N_2_-fixing bacterium *Azotobacter chroococcum* provides N to the alga *Chlamydomonas* and promotes its growth, whereas *Chlamydomonas* fixes CO_2_ and provides organic carbon to the bacterium. This symbiotic interaction has reported benefits for the industry by increasing lipid production in the microalga [8].

Anthropomorphic activities have affected the natural equilibrium of the N cycle for more than a century. The industrial N_2_-fixation into ammonia (Haber-Bosch process) and the use of this ammonia as fertilizers has dramatically improved crop productivity but, at the same time, it has increased the reactive N load by more than 120%, which includes all biologically active, photochemically reactive, and radiatively active N compounds in the atmosphere and biosphere [1,2,9,10]. This increase in the N load comprises a higher global amount of reduced inorganic forms of N (NH_3_ and NH_4_^+^), oxidized inorganic forms (as NOx, HNO_3_, N_2_O, and NO_3_^−^), and organic compounds (such as urea, amines, and proteins), in detriment to the unreactive N_2_ gas. One of the main unwanted effects of this imbalance in the N cycle is the transfer of reactive N from terrestrial to coastal systems and ground waters, causing algal blooms and the eutrophication of these ecological niches [9,10]. In addition, it is estimated that 3–5% of the N used in fertilizers is converted into N_2_O, a greenhouse gas that is 310 times more potent than CO_2_ and considered the dominant ozone-depleting pollutant emitted in the 21st century [1,11,12,13]. The principal substrate for N_2_O synthesis is NO, which is mainly reduced by bacterial NO reductases but also can be reduced by some eukaryotic enzymes that carry out nitrate dissimilatory reactions. 

Nitrogen is a macronutrient for plants, algae, and bacterial growth. Nitrate is the most oxidized form of nitrogen and highly abundant in aquatic habitats. The nitrate assimilation pathway (NO_3_^−^ → NO_2_^−^ → NH_4_^+^ → amino acids) is quantitatively the most important redox process, supporting almost 20% of marine algal growth [1,4]. Dissimilatory nitrate reduction pathways such as dissimilatory nitrate reduction to ammonium (DNRA), denitrification to N_2_O, and total denitrification to N_2_ are also processes mediated by eukaryotes in aquatic systems. DNRA is described in diatoms as an adaptation and survival process to darkness and anoxia, where the intracellular NO_3_^−^ is used as an electron acceptor instead of O_2_ [14,15]. On the other hand, a complete denitrification process (NO_3_^−^ → NO_2_^−^ → NO → N_2_O → N_2_) has been described in some species belonging to the foraminifera group, which represent the only eukaryotic organisms able to perform complete denitrification to N_2_ [16,17,18]. Other foraminifer species perform only partial nitrate denitrification to N_2_O [17,18,19]. The genes involved in the foraminifer denitrification pathway have been recently identified [20]; however, the genes encoding the enzymes responsible for the first and last steps of this pathway, the dissimilatory nitrate reductases and nitrous oxide reductase, are missing in most of the foraminifer genomes but found in the associated bacteria. Partial nitrate denitrification to N_2_O has also been well documented in fungi [21,22,23,24,25] and recently described in the green microalga *Chlamydomonas reinhardtii,* where a NO reductase (CYP55) [26] and flavodiiron proteins (FLVs) [27] catalyze the reduction of NO to N_2_O in dark and light conditions, respectively. These two proteins constitute the molecular evidence that supports that microalgae contribute to N_2_O emissions, something known for decades [28]. In this review, we analyze the metabolic flexibility of microalgae to carry out different steps of the N cycle with special attention to the denitrification pathway. 

## 2. The Role of *Chlamydomonas reinhardtii* in the N Cycle

Traditionally the genus *Chlamydomonas* corresponds to unicellular, biflagellate, green microalgae and encompasses more than 500 species. However, current phylogenetic studies have reevaluated the taxonomy of *Chlamydomonas* into three species: *C. reinhardtii*, *C. incerta,* and *C. schloesseri* [29,30]. Today, all the unequivocally identified *C. reinhardtii* cells have been isolated from soil habitats suggesting a preference for rich nutrient environments. *Chlamydomonas* spp., other than *C. reinhardtii,* have been isolated from very different environments such as acid mine drainage, temperate marine, Antarctic ice, and even air [29]. *C. reinhardtii*, named as *Chlamydomonas* throughout, has been developed as a powerful model organism useful to study essential biological processes but also biotechnological applications [31].

Nitrogen assimilation has been widely studied in *Chlamydomonas* and the pathways for the assimilation of different forms of nitrogen, inorganic (nitrate, nitrite, and ammonium) as well as organic (urea, amino acids, and purines), have been identified [32,33,34]. However, besides the classical N assimilation pathways that have been described in axenic *Chlamydomonas* cultures, recent advances on nitrogen-transforming reactions have revealed that this microalga can also carry out both an ammonification process by mutualistic interactions with bacteria and a denitrification process that releases N_2_O to the environment (Figure 1, Table 1). 

### 2.1. Ammonification and Mutualism with Bacteria for N Scavenging from Amino Acids and Peptides

*Chlamydomonas* cells scavenge nitrogen from amino acids in three ways: transport, deamination/ammonification, and mutualistic interaction with bacteria [41]. Arginine is the only amino acid that is efficiently transported into the cell from the surrounding media and then used as carbon and nitrogen sources [42]. L-leucine and L-cysteine is also transported, but by passive diffusion [43]. However, most of the proteinogenic amino acids, including arginine, but also di- and tri-peptides, can be extracellularly deaminated, producing NH_4_^+^ and the corresponding keto acids. This ammonification process is mediated by the periplasmic L-amino acid oxidase (LAO1) (step 7, Figure 1, and Table 1), which is induced upon inorganic N deprivation [41,44,45,46,47]. Interestingly, only the NH_4_^+^ generated by the LAO1 enzyme is transported inside the cell by high-affinity ammonium transporters (AMT) and used by the alga, but the carbon skeletons remain in the medium [47]. It has been hypothesized that such carbon skeletons could be used for establishing mutualistic interactions with specific bacteria in the surrounding medium. These interactions depend on whether the released keto acid can meet the particular carbon requirements of a specific bacterium [41]. Besides ammonium and the keto acid, the ammonification process mediated by LAO1 also generates H_2_O_2_, which has bactericidal properties [48]. The eco-physiological role of H_2_O_2_, if any, is not clear but its production could repel other microorganisms to avoid competition for the organic N and, simultaneously, promote bacterial cell death making more N available. Alternatively, H_2_O_2_ synthesis could also serve as a mechanism to select specific H_2_O_2_-tolerant bacterial species in order to restrict the mutualistic interactions. The role of H_2_O_2_ in algal-bacterial mutualistic interaction has been previously reported [49].

*Chlamydomonas* LAO1 homologs have been found in other algal species, including Rhodophyta, Alveolata, Heterokonta, Haptophyta, and Dinophyta species, but not in other chlorophytes [41].

Despite the promiscuous LAO1 activity to deaminate amino acids and oligopeptides, Calatrava and collaborators [50] have reported that this enzyme cannot use proline and some oligopeptides as substrate. Interestingly, when proline is the only N source in the media, *Chlamydomonas* can establish a strict and specific mutualism with Methylobacterium species [50]. Under this situation, the bacteria mineralize the amino acid (or peptide) releasing ammonium that can be incorporated by the alga. In return, *Chlamydomonas*, which fixes CO_2_ to satisfy its carbon demand, produces C-compounds as glycerol that can be secreted and used as a C source for the bacterium.

Another step in which *Chlamydomonas* could participate in the N cycle is the use of N_2_, as described from co-cultures with the nitrogen-fixing anaerobic bacteria *Azotobacter chroococcum.* In this situation, the bacteria supply the nitrogen source and CO_2_ to the alga by nitrogen fixation, meanwhile, the alga supplies carbohydrates and O_2_ via photosynthesis to the bacteria [8].

### 2.2. Nitrate Denitrification

The nitrate flux toward its assimilation is the primary nitrogen pathway that occurs in *Chlamydomonas* as well as in eukaryotic organisms containing an assimilatory nitrate reductase (NR). The nitrate assimilation appears to occur via a relatively simple linear pathway involving (1) nitrate uptake by specific nitrate transporters (i.e., NRT2.1/NAR2), (2) cytosolic nitrate reduction by a typical eukaryotic nitrate reductase (NR), (3) nitrite transport into the chloroplast by specific transporters (i.e., NAR1.1), (4) nitrite reduction to ammonium by a typical plant-like ferredoxin-dependent nitrite reductase (NIR), and finally (5) the ammonium incorporation into carbon skeletons by the GS/GOGAT cycle that synthesizes glutamate [33,34].

Besides its assimilatory role, the NR is a moonlighting protein that interacts with different partners to carry out alternative reactions. In photosynthetic organisms, NR is considered one of the principal nitric oxide (NO) sources [35] and also participates, as shown in *Chlamydomonas*, in the NO oxygenation to convert this signaling molecule back to nitrate. These reactions, in which NR is central and acts with different partners, can be called the NO_3_^-^/NO_3_^-^ cycle (steps 1, 4, and 5 in Figure 1). Moreover, the intracellular NO can also be the substrate for NO reductases producing N_2_O in mitochondria and chloroplasts [1]. Thus, NR seems to be a cross-point among three different pathways: nitrate assimilation, a partial nitrate-denitrification to NO and the contaminant gas N_2_O, and the NO oxygenation to nitrate.

#### 2.2.1. The NO_3_^−^/NO_3_^−^ Cycle

The eukaryotic NR is a homo-dimeric protein that contains three prosthetic groups in each subunit (FAD, Heme b_557_, and molybdenum cofactor –Moco-). Nitrate reduction to nitrite is a redox reaction that occurs across a mini electron transport chain (mini-ETC) from NAD(P)H to nitrate through three prosthetic groups (NAD(P)H → FAD → heme → Moco → NO_3_^−^). However, as mentioned above, NR can bind to other protein partners and detour electrons at different levels of this mini-ETC. When the partner protein is the molybdoprotein NOFNIR (NO forming nitrite reductase), electrons flow until the heme cofactor, where they reduce the NOFNIR that converts nitrite to NO [35,51]. When the partner is the truncated hemoglobin 1 (THB1), electrons go out of the mini-ETC at the FAD cofactor level and reduce the THB1 protein, which transforms NO into nitrate [33]. These three reactions constitute the named NO_3_^−^/NO_3_^−^ cycle. 

Nitric oxide is a signal molecule that regulates a high number of physiological processes in mammals, plants, and algae [52]. In *Chlamydomonas*, NO regulates N assimilation at the transcriptional and posttranslational level. NO is a negative regulator for the expression and activities of the high-affinity nitrate/nitrite transporters, the NR, and the high-affinity ammonium transporters [33,53]. As a signal molecule, NO is expected to have a temporal effect, and therefore, its synthesis and degradation have to be tightly controlled. 

This NO_3_^−^/NO_3_^−^ cycle could work for regulating nitrate assimilation without N loss. When the conditions are optimal, nitrate goes toward its assimilation up to glutamate. If the conditions are not optimal (i.e., α-ketoglutarate is deficient) and nitrite accumulates, NO is synthesized and stops the nitrate uptake and reduction. However, although NO is a gas and can diffuse out of the cells, the NR-THB complex could convert the NO into nitrate, avoiding the N loss. 

#### 2.2.2. NO Synthesis and Reduction: A Prokaryote Pathway 

In plants and plant-like organisms, NO is produced by different mechanisms that use different substrates (i.e., NO_2_^−^ or arginine) in different intracellular compartments such as cytosol [35,54,55], mitochondria [26,56], chloroplast [57,58], and peroxisomes [59]. The oxidation of arginine to NO and citrulline by the NO synthase (NOS) is a well-known reaction in mammals, bacteria, and some algae [60,61]. NOS-like reactions have been described in land plants (peroxisomes) [62] and *Chlamydomonas* [63], but extensive genome analyses have ruled out the existence of NOS in photosynthetic organisms except for some microalgae [60,61]. However, NO production from NO_2_^−^ is well documented to occur in the cytosol, by NR [54,55] or NR/NOFNIR [35], and also in mitochondria [26,56] and chloroplast by the electron transport chains [57]. 

Besides NO conversion into nitrate, NO can also be reduced by NO reductases to N_2_O. This denitrification process implies the loss of nitrogen to the atmosphere as a potent greenhouse gas. This pathway (NO_2_^−^ → NO → N_2_O) has been well defined in both prokaryotes [1] and eukaryotes (fungi). In fungi, denitrification involves NIRK (copper-containing nitrite reductase) (NO_2_^−^ → NO) and the cytochrome P450, CYP55 (NO → N_2_O), both acquired from bacteria [22]. For fungal CYP55, a horizontal gene transfer from bacteria is suggested, but for NIRK, a proto-mitochondrial origin has been proposed.

The *Chlamydomonas* genome contains two genes encoding for both proteins, NIRK and CYP55 (Table 1). CYP55 appears to be responsible for N_2_O production in the dark [26,27]. Accordingly, CYP55 is located in mitochondria [38] and its transcript is accumulated in the dark [64] and in the presence of nitrate [38]. The dark NO reduction attributed to CYP55 activity has also been shown to be stronger when cells are grown in the presence of nitrate [27]. Although found in the mitochondrial proteome, CYP55 is also predicted to be targeted to the chloroplast [27]. Thus, it is possible that the protein is targeted to both organelles. In this case, CYP55 could be able to use electron donors from photosynthetic origin. Like CYP55, NIRK is also up-regulated in the dark [64] and in nitrate-containing media (Calatrava, unpublished data). In silico analyses strongly suggest that NIRK could also be localized in the mitochondria (Table 1), pointing out that *Chlamydomonas* seems to have a complete and operative NO_2_^−^ → NO → N_2_O pathway in the mitochondria.

The second type of protein shown to reduce NO to N_2_O, in *Chlamydomonas*, corresponds to FLVs (flavodiiron proteins) [27]. Flavodiiron proteins belong to a singular family of oxygen and/or nitric oxide reductases widespread in prokaryote and eukaryote organisms. This protein family has been classified into eight classes (A to H) according to the presence of modular domains [65,66]. Class A FDPs are the most dominant class in the family and only contain the two core domains (Fe-Fe and FMN) [65,66]. Some examples for this class A show a preference for O_2_ as revealed for the protozoal and archaeal enzyme [67,68,69,70], or they may have a dual activity toward NO and O_2_ as in *M. thermoacetica* or *D. gigas* [71,72]. Class B FDPs contain an additional C-terminal rubredoxin domain and, because of that, were named flavorubredoxins (FIRD) or NORV (from the encoding gene in *E. coli*) [65]. In *E. coli*, FLVB/FIRD/NORV was shown to have significant preference for NO over O_2_ and confers resistance to NO under anaerobic conditions [73,74]. Class C FDPs have a C-terminal flavin-binding domain, predicted to have a flavin reductases activity accepting electrons from NAD(P)H [65,66]. This class C is present in cyanobacteria, the protozoan *Paulinella chromatophora*, green algae, mosses, lycophytes, and gymnosperms, but absent in angiosperms [65]. *Chlamydomonas* FDPs correspond to class C (also called Flv), which forms a distinct cluster that falls into two groups. One of these groups (FlvA) has the ligand binding diiron site strictly conserved, while in the other (FlvB) the ligand binding sites are not conserved. The *Chlamydomonas* genome contains two genes, *FLVA* and *FLVB*, encoding both groups of flavodiiron proteins. Moreover, the *Chlamydomonas* FLVs could work as a heterodimer as shown by different insertional mutants of the *FLVB* gene that result in the loss of expression of both FLVB and FLVA proteins [71]. The study of these mutants revealed that FLVs participate in a Mehler-like reduction of O_2_, providing an alternative valve for electrons during the dark to light transition in the chloroplast. Thus, FLVs are critical for the algal growth under fluctuating light regimes but not in continuous light [75]. These mutants were also useful to prove the FLVs role as a NO reductase, being the major enzyme responsible for NO reduction to N_2_O under conditions of light in the presence of O_2_ in *Chlamydomonas*. In addition, in vivo experiments suggest that *Chlamydomonas* FLVs shows a higher affinity for NO than for O_2_ [27]. Therefore, *Chlamydomonas* has inherited two types of N_2_O-producing proteins, FLVs in the chloroplast and CYP55 in the mitochondria.

Besides *CYP55* and *FLVs*, the *Chlamydomonas* genome also contains four genes encoding for hybrid cluster proteins (HCPs) [39], which are metalloproteins characterized by the presence of an iron-sulfur-oxygen cluster considered to be unique in biology [25]. The analysis of these protein sequences and their phylogenetic analyses support that the HCP family has emerged from a single gene of an alpha-proteobacterium with subsequent duplications in *Chlamydomonas* [39]. Like for CYP55 and NIRK, both nitrate and darkness contribute to the strong up-regulation of the *HCPs* in *Chlamydomonas* growing under oxic conditions [39]. In dark anoxia, the *HCP4* transcript shows the stronger induction [76], and the studies performed in mutants obtained by gene silencing suggest a role in regulating fermentative and nitrogen metabolism pathways [77]. Concerning their cellular localization, HCP1, 3, and 4 are predicted to be in the chloroplast [39], but HCP2 is located in the mitochondria, as confirmed in mito-proteome analysis [38]. In bacteria, the function(s) of HCP is debated. Initially, they were considered hydroxylamine reductases, as they can use this substrate to produce ammonium in vitro, but the low affinity for hydroxylamine has questioned this physiological role [78]. Later, Wang and collaborators demonstrated that *E. Coli* HCP is a high-affinity nitric oxide (NO) reductase and proposed it as the major enzyme for reducing NO to N_2_O under physiologically relevant conditions [25]. The in vivo high affinity NOR activity is also demonstrated in *Methylobacter tundripaludum*, where NO mediates a response to hypoxia, quorum sensing, the secondary metabolite tundrenone, and methanol dehydrogenase [79]. What makes the history of HCP attractive is that it can function as a protein S-nitrosylase with a role in the NO sensing pathway. Thus, HCP is auto-S-nitrosylated and afterward results in the assembly of an HCP interactome, including enzymes that generate NO, synthesize SNO-proteins, and propagate SNO-based signaling mediating the NO-mediated stress responses [80]. The function(s) of HCP in *Chlamydomonas* as NOR and or protein-S-nitrosylase has to be solved as well as its relationship with nitrate metabolism. Seth and collaborators propose that HCPs oxidize NO to NO^+^, which is the form required for S-nitrosylation [80]. However, the redox potentials of both HCP and HCR (HCP reductase, also required for HCP function) do not fit with an oxidase activity and Lis et al. have proposed that HCPs might exhibit reductase activity and perform the following two-electron reduction reaction, NO_3_^−^ + 4H^+^ + 2e^−^ → NO^+^ + 2H_2_O [39].

## 3. N_2_O Production and Enzymes in Other Microalgae

For decades, N_2_O emission has been detected from lakes, coasts, and oceans, ecosystems where microalgae are mainly present [81], although the contribution of microalgae to these N_2_O emissions has been mostly neglected and attributed to the prokaryotic organisms present in these environments [28,82,83,84]. However, the correlation between algal bloom events, triggered by the excess of N from anthropogenic origins, and N_2_O emissions suggests that these microorganisms may be partly responsible for this production [1,28,83]. Importantly, many microalgae are currently cultivated at a large scale for commercial and biotechnological purposes. Therefore, the potential contribution to N_2_O emissions of these microalgae should be considered [26,28]. Microalgae cultures, supplied with NO_3_**^−^** as the sole N source, have been estimated to emit N_2_O in amounts 2.5 to 14 times more than from natural vegetation [85,86]. Although the emission of N_2_O by microalgae depends on many environmental factors, the major ones seem to be the species of microalga and the N source [26,27,87]. While N_2_O is produced on nitrate or nitrite, its production is prevented on ammonium [28].

The existence of a full NO_2_^−^ → N_2_O dissimilation pathway in *Chlamydomonas* made interesting the analysis of other microalgae genomes. Public algal databases were searched for homologous proteins to the *Chlamydomonas* NIRK, possibly involved in NO synthesis and CYP55, FLVs, HCP as potential NO reductases. Those microalgae containing at least one of these proteins were selected. A total of 33 microalgae were considered, 23 Chlorophyta and 10 non-Chlorophyta species. Considering there are around one hundred microalgae genomes available, it can be assumed that almost 33% of microalgae contain at least one of these proteins. Phylogenetic trees of these proteins were constructed to provide a broader picture of the distribution of these proteins in the tree of life consistent with previous reports for FLV [65] (Appendix A), HCP [39] (Appendix A), CYP55 [88] (Appendix A), and NIRK [21] (Appendix A). These data were correlated to the production of N_2_O by the microalgae (Table 2) and discussed.

From the 33 selected microalgae, only three Chlorophyta species (*Chlamydomonas reinhardii,* two subspecies of *Scenedesmus obliquus,* and *Chlorella sorokiniana)* contained a full set NIRK, CYP55, FLVA, FLVB, and HCP. In other words, they contain a prokaryote NO synthetase and redundant potential NO reductases. This represents almost 9% of the microalgae in Table 2. In addition, these three microalgae have been reported to produce N_2_O [26,27,89].

For the rest of microalgae in Table 2, an incomplete NO_2_^−^ → N_2_O pathway or the potential to produce NO (NIRK) or reduce NO to N_2_O through FLVs, CYP55, or HCP reductases was observed. The molecular evidence for these NO reductases is restricted to a few microalgae and comes from a recent study by Burlacot and collaborators [27]. Both CYP55 and FLVs were responsible for producing N_2_O under two different conditions, CYP55 in the dark and FLVs in the light. Therefore, microalga strains lacking CYP55 and/or FLV also are unable to produce N_2_O under its particular condition. However, whether HCP is or is not a NO reductase is unsolved in microalgae.

As previously reported, FLVs are present only in Chlorophyta and in approximately 74% of the selected chlorophytes in Table 2. These species contain both FLVA and FLVB, suggesting that these proteins may function as a heterodimer [75]. The phylogenetic tree suggests that FLVs from chlorophytes and early-diverged land plants are related to cyanobacteria (Appendix A), most likely derived from the endosymbiotic gene-transfer (EGT) of both genes from the chloroplast to the nuclear genome in the green lineage, after the divergence of the red algae and glaucophytes. Otherwise, both genes may have been transferred to the nucleus of the ancestral archaeplastidan and subsequently lost in red algae and glaucophytes. 

As previously reported, *HCP* genes, although not present land plants, are highly represented in Chlorophyta; most of the chlorophytes harbor more than one copy of this gene [39]. In *Chlamydomonas reinhardtii,* the four *HCP* gene copies are phylogenetically related and located in the same chromosome, which suggests that these genes were recently duplicated [39]. In Table 2 it is shown that 69% of the chlorophytes harbor HCP. Moreover, putative HCP orthologs are also found in nonchlorophytes, including three *Nannochloropsis* spp. (Eustigmatales), the *Guillardia thetha* and the Stramenopiles *Phaeodactylum tricornutum* and *Thalassiosira pseudonana* (Table 2, Appendix A). Supported by further phylogenetic analyses, an endosymbiotic origin of the microalgal HCP genes from mitochondria has been proposed [39].

Although less abundant, CYP55 is in almost 39% of the Chlorophyta in Table 2 and Appendix A. It was also found only in one of other non-chlorophytes (*Thalassiosira pseudonana*), but it corresponds to a truncated sequence and it was not included in the phylogenetic tree. The most closely related to microalgal CYP55 are fungal members, which are hypothesized to have been acquired from Actinobacteria [22]. In fungi, CYP55, known as p450nor (cytochrome p450 NO reductase), is considered as a biomarker for N_2_O production, since it is detected in all fungal isolates producing N_2_O from NO_2_^−^ [24]. Because *Actinobacterial* are not considered canonical denitrifying bacteria, the prevalent hypothesis for the origin and evolution of CYP55/p450nor is its acquisition from *Actinobacteria* and next evolution to a new role in denitrification [22]. Recently, a new hypothesis suggests that fungal p450nor has a role in secondary metabolism rather than denitrification and that N_2_O is a minor bioproduct of the secondary metabolism [88].

The NO-forming NIRK, which shares a common bacterial origin in both fungi and microalgae [21], is poorly represented in chlorophytes (3 out of 33) (Table 2, Appendix A), but it is present in the genome of the red alga *Galdieria sulpharia,* the haptophyte *Emiliana huxleyi*, and in Excavata. However, the NO reductase activity by this protein in these species is still unknown.

## 4. Concluding Remarks and Future Perspectives

N homeostasis in ecosystems relies on the balance between N fixation and N release to the atmosphere in gaseous forms (mainly N_2_ and, to a lower extent, N_2_O and NO). In-depth studies of the reactions mediated by isolated organisms, as well as those carried out by consortiums, will help us to understand how organisms keep the equilibrium of the biogeochemical N cycle and will allow us to forecast putative imbalances.

*Chlamydomonas reinhardtii* performs a high number of N-transforming reactions. Quantitatively and preferentially are those devoted to inorganic nitrogen (NH_4_^+^, NO_3_^−^, NO_3_^−^) assimilation for biomass production. The ammonification/deamination of amino acids and peptides constitutes a poor N supply but leads to bacterial mutualistic interactions for C/N exchange. NO_3_^−^ offers additional N-transforming reactions, such as the NO_3_^−^/NO_3_^−^ cycle, and dissimilation through a typical prokaryotic pathway NO_2_^−^ → NO → N_2_O by means of apparently redundant NO reductases.

Algae, playing essential roles in aquatic ecosystems in carbon an N cycles, were claimed for a long time to produce the potent greenhouse gas N_2_O. This point was recently further confirmed by molecular evidence with CYP55 and FLVs in *Chlamydomonas reinhardtii*. This review opens new perspectives for future investigations of the full or partial dissimilation pathway (NO_2_^−^→ NO → N_2_O) by algae, where NIRK, CYP55, FLVs, and HCPs may be involved. In fact, some green microalgae seem to share the dissimilating full pathway NO_2_^−^ → NO → N_2_O with *Chlamydomonas*. However, partial pathways seem to be prevalent in microalgae. 

Understanding this NO_2_^−^ → NO → N_2_O pathway operating in microalgae as complete or partial routes will be important to understand the production and regulation of either a signal molecule as is NO or the loss of nitrogen as the contaminant gas N_2_O.

## Figures and Tables

**Figure 1 plants-09-00903-f001:**
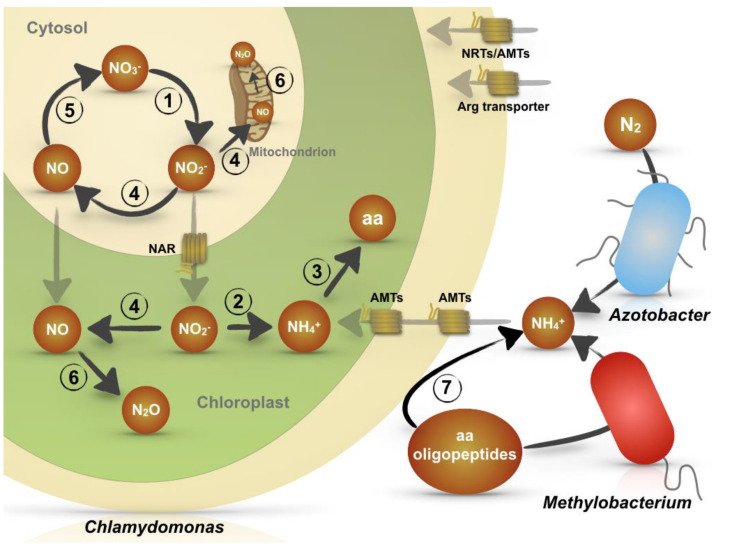
Nitrogen-transforming reactions by *Chlamydomonas reinhardtii.* NO_3_^−^ assimilation (steps 1,2,3); NO_3_^−^/NO_3_^−^ cycle, (steps 1,4,5); NO_3_^−^ dissimilation (steps 4,6) and ammonification from amino acids and peptides (step 7). Transporters for ammonium (AMT), nitrate (NRT), and arginine are indicated. Mutualism with bacteria for N/C exchange is also shown.

**Table 1 plants-09-00903-t001:** Enzymes for nitrogen-transformation reactions in *Chlamydomonas*.

Step		Enzyme	Gene ID	Regulation	Localization	Ref.
**1**	NO_3_^–^ + 2e^–^ + 2H^+^→ NO_2_^–^ + H_2_O	NR	Cre09.g410950	NO_3_^–^	Cytosol	[33]
**2**	NO_2_^–^ + 6e^–^ + 8H^+^ → NH_4_^+^ + 2H_2_O	NIR	Cre09.g410950	NO_3_^–^		[33]
**3**	NH_4_^+^ + 2e^–^ + α-ketoglutarate + ATP→ Glutamate + ADP + Pi	GS/GOGAT				[33]
**4**	NO_2_^–^ + e^–^ + 2H^+^ → NO + H_2_O	NR/NOFNIR NirK	Cre09.g395950/Cre09.g389089 Cre08.g360550	NO_3_^–^ Dark	Cytosol Mitochondria	[35] [36]
**5**	NO + 2H_2_O → NO_3_^–^ + 3e^–^ + 4H^+^	NR/THB1	Cre09.g410950/Cre14.g615400	NO_3_^–^	Cytosol	[37]
**6**	2NO + 2e^–^ + 2H^+^ → N_2_O + H_2_O	CYP55 FLVA FLVB HCP1 HCP2 HCP3 HCP4	Cre01.g007950 Cre12.g531900 Cre16.g691800 Cre09.g391650 Cre09.g393543 Cre09.g393506 Cre09.g391450	NO_3_^–^/Dark - - NO_3_^–^/Dark NO_3_^–^ NO_3_^–^/Dark NO_3_^–^/Dark/anaerobiosis	Mitochondria* Chloroplast Chloroplast Chloroplast Mitochondria* Chloroplast Chloroplast	[38] [27] [27] [39] [38] [39] [40]
**7**	L-Amino acid +H_2_O+ O_2_ → NH_4_^+^ + α-keto acid + H_2_O_2_	LAO1	Cre12.g551353	Nitrogen starvation	Periplasm	[35]

Steps 1 to 7 correspond to those in Figure 1. Gene identification (Phytozome). * Protein indentified in the mitoproteome. NR (Nitrate Reductase), NIR (Nitrite Reductase), GS (Glutamine Synthetase), GOGAT (Glutamine Oxoglutarate Aminotransferase), NOFNIR (NO Forming Nitrite Reductase), NirK (Copper-Containing Nitrite Reductase), THB1 (Truncated Hemoglobin 1), CYP55 (Cytochrome P450), FLV (Flavodiiron Proteins), HCP (Hybrid Cluster Protein) and LAO1 (L-Amino Acid Oxidase).

**Table 2 plants-09-00903-t002:** Microalgal production of N2O and putative NIRK proteins, involved in NO formation, and putative HCP, CYP55, and FLV, involved in NO reduction to N_2_O.

Microalgae	NIRK	HCP	CYP55	FLVA	FLVB	N_2_O/Ref.
*Chlamydomonas reinhardtii*	PNW79625.1	XP_001694756.1 XP_001694571.1 XP_001694671.1 XP_001694454.1	XP_001700272.1	XP_001692916.1	PNW71243.1	+ [26,27]
*Chlamydomonas eustigma*		GAX74118.1		GAX73888.1	GAX73438.1	
*Chlorella variabilis*		XP_005851548.1	XP_005848845.1	XP 005849742.1	XP 005849474.1	+ [27]
*Chlorella sorokiniana*	PRW57688.1	PRW58172.1	PRW58571.1 (A)	PRW59370.1	PRW58882.1	
		PRW60486.1	PRW58572.1 (B)			
		PRW58278.1				
*Scenedesmus obliquus UTEX 393*	SZX67391.1	ID:15469	ID:435	SZX62872.1	SZX63966.1	+ [89]
		ID:3042		SZX76545.1	SZX639665.1	
*Scenedesmus obliquus EN0004*	ID:579459	ID:571596	ID:324854	ID:321778/332768	XP002948938.1	+ [89]
		ID:618262		ID:342393		
*Nannochloropsis salina*		NSK 004142				+ [90,91]
*Nannochloropsis oceanica CCMP1779*		599317				
*Nannochloropsis gaditana*		XP 005853562.1				− [27]
*Gonium pectorale*		KXZ41847.1	KXZ42279.1	KXZ51271.1	KXZ42718.1	
		KXZ44628.1				
*Coccomyxa subelipsoidea*				XP 005643451.1	XP 005643449.1	+ [27]
*Tetrabaena socialis*		PNH09786.1				
*Raphidocelis subcapitata*		GBF99305.1	GBF95475.1	GBF88523.1	GBF98453.1	
		GBF96263.1				
*Volvox carteri*		XP_002952683.1		XP 002947563.1	XP 002948938.1	
*Yamagishiella unicocca*		ARO50069.1				
*Micractinium conductrix*		PSC73563.1		PSC76351.1	PSC76929.1	
		PSC72373.1				
		PSC68490.1				
*Micromonas pusilla*				XP 003057013.1	XP 003057495.1	
*Micromonas commoda*				XP 002502419.1	XP 002502038.1	
*Monoraphidium neglectum SAG 48.87*		XP013900867.1	ID:2893 ID:2894			
*Bathycoccus prasinos*				XP007514084.1	XP 007514099.1	
*Bathycoccus lucimarinus*				XP001416100.1	XP001416098.1	
*Ostreococcus tauri*				XP003075211.2	XP003075215.2	
*Klebsormidium nitens*				GAQ79906.1	GAQ89021.1	
*Galdieria sulphuraria*	XP_005707361.1					− [27]
*Porphyridium purpureum*						− [27]
*Phaeodactylum tricornutum*		ID:12416				− [27]
*Thalassiosira pseudonana*		ID:32716	ID:263399^a^			− [27]
*Guillardia theta CCMP2712*		ID:152637				
*Andalucia godoyi*	KAF0852161.1					
*Phaeocystis antarctica CCMP1374*	ID:684					
*Emiliana huxleyi*	XP 005783766.1					

Reciprocal BLAST was performed using the *Chlamydomonas* homolog of each protein as a query. Green background highlights microalgal species that belong to the Chlorophyta phylum. For those algal species that did not show in a positive hit using the current available genome sequences, the gene coding for the corresponding queried protein was considered to be absent. Protein accession numbers correspond to those indicated in Appendix A, from NCBI or Phycocosm (protein references obtained from Phycocosm are indicated as follows “ID: (ref number)”. “^a^” indicates partial sequence. N_2_O/Ref., positive or negative (+/−) N_2_O production by the microalgae and empty when it is unknown.

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
