# Peer review of "Chlamydomonas reinhardtii, an Algal Model in the Nitrogen Cycle"

_plants, 2020, doi:10.3390/plants9070903_

Round 1

Reviewer 1 Report

The problem presented in the manuscript entitled “Chlamydomonas reinhardtii, an algal model in the nitrogen cycle” is very interesting. However, some aspects of the work need to be explained or improved.

  1. 37: The expression “In nature” is doubled in the sentence.
  2. 38-39: “These N reservoirs and their fluxes” – I suppose Authors did mean “N flux” not “reservoirs fluxes”.
  3. 71-76: Here, I would recommend refining the text fragment, because two types UCYN-A are known; one of them is associated to Braarudosphaera bigelowii, and the second one is associated to relatives of B. bigelowii.
  4. 93: I would say that N, rather than nitrate is a macronutrient.
  5. 119-120: Here, I would recommend refining the text fragment, because not all C. reinhardtii strains originate from this particular strain isolated in 1945. This strain is, however, the base for many laboratory strains.
  6. 140-142: Indeed, arginine is the only known amino acid that is actively transported into Chlamydomonas cell. However, L-leucine and L-cysteine is also transported, but by passive diffusion. Please, add the appropriate information.
  7. 151-156: Despite animal LAOs are reported to have antibacterial properties, such a function was not described for algae. Assuming, that algal LAO is an element of alga-bacteria mutualism, antibacterial action of the enzyme is rather unlikely. Could you consider other explanation, e.g. signaling role of H2O2?
  8. 296: “s” is missing in “Chlamydomonas”.
  9. Table 2: The heading is confusing. Please, rewrite it to make it more informative.
  10. Please, check the reference list, since I have found some inconsistencies, e.g. ref. 5: the journal name is incorrect; ref. 29: the title is incomplete, etc.

Author Response

Please see the PDF attached

Reviewer 2 Report

The review by Bellido-Pedraza et al. describes the current knowledge related to the nitrogen cycle in microalgae and provides a valuable overview of the current literature. This area of research is of utmost importance since the nitrogen metabolism of microalgae is involved in N2O production, a potent greenhouse gas 300 times more powerful than CO2. The review is well written with clear focus and good references.

My main concerns relates to the part dealing with Flavodiiron proteins, which is misleading and should be amended (line 235 to 243). There is here a misunderstanding of the FLVs classification. Although some authors have classified FLVs from A to H based on the presence or absence of some accessory domains in the proteins, this has nothing to do with the naming of FLVA and FLVB found in Eukaryotes. Eukaryotic FLVA and FLVB both belong to the Class C of the flavodiiron proteins as described by Romao et al, 2016 (cited in the review) and Folgosa et al, FEMS letter 2017 (not cited in the review). Please consider revising the discussion here.

Another main concern is related to the fact that the concluding remarks are quite poor and should be improved by making them more attractive and adding future perspectives.

More minor remarks:

Line 37: “In nature, living organisms can use different N reservoirs in nature,”, please remove one “in nature”

Lines 62-65: This sentence does not fit well in the introduction. Should better fit as a perspective.

Line 68: Please provide also examples and references of enzymes of the nitrogen metalolism only found in eukaryotes.    

Line 74: “In the lab, the N2-fixing bacterium Azotobacter chroococcum provides N to the alga and promotes its growth, whereas Chlamydomonas fixes CO2” The phrasing of this phrase is misleading: when reading, “the alga” one can refer to the algal called in the previous sentence (Braarudosphaera bigelowii). Please reformulate.

Line 76: It is not clear how supplying nitrogen to a Chlamydomonas culture increases lipid production since most studies have shown that lipid production is enhanced by nitrogen deprivation. Please comment on this point.

Line 109: “a flavodiiron protein (FLV)“ should be modified since there are two Flavodiiron proteins in Chlamydomonas.

Figure 1: Please provide in the legend the description of the different transporters shown in the figure.

Table 1: for the LAO1 regulation, it may be more explicit to write “Nitrogen starvation” or “Nitrogen deprivation” instead of “Minus Nitrogen”

Line 212: The sentence is not clear. “NO synthesis is produced by different mechanisms” should better read “NO is produced by different mechanisms. Please explain what are alternative substrates?

Line 222: Could you please provide the full name of the NIRK?

Line 227: “Accordingly, CYP55 is located in mitochondria” Although found in the mitochondrial proteome, CYP55 is predicted to be targeted to the chloroplast (Burlacot et al, 2020). Maybe this discrepancy could be discussed here. It is possible that the protein is targeted to both organelles, especially since CYP55 is be able to use electron donors from photosynthetic origin.

Line 228: The dark NO reduction attributed to CYP55 activity has also been shown to be stronger when cells are grown in the presence of nitrate (Burlacot et al, 2020).

Line 246: The demonstration that FLVs participate in O2 photoreduction as an alternative electron flow has been made by Chaux et al, 2017, not by Saroussi et al, 2019. The work by Saroussi et al, 2019 is focused on the interaction with starch metabolism, which is beyond the focus of the current review.

Line 295: Please provide a reference for “While N2O is produced on nitrate or nitrite, its production is prevented on ammonium. “

Line 298: Why the authors did not consider NOFNIR, which has also been proposed to produce NO? From previous work published by some of the authors, NOFNIR seems to be important in NO production in Chlamydomonas (Chamizo-Ampudia et al, 2016). Please justify.

Table 2: What criteria was used to define the presence or absence of one of the proteins?

General remark: Current data from Plouviez et al, 2017 and Burlacot et al, 2020 suggest that in Chlamydomonas, CYP55 and FLVs are the main contributors to N2O production. Why do the authors favor the hypothesis of HCPs being NOR rather than being a reductase that would be involved in other reactions of the N-cycle, like the reduction of nitrates to NO+? Could you please argue on this point?

Author Response

Please see the PDF attached
